# Learning Task-Invariant Features in VLMs via Dynamic Bayesian IRM

## Abstract

While Visual Language Models (VLMs) excel on multimodal tasks, they suffer from performance degradation under distribution shift, particularly when facing out-of-distribution (OOD) tasks not seen during training. Traditional Empirical Risk Minimization (ERM) often fails to learn task-invariant features, relying instead on spurious correlations. Although Invariant Risk Minimization (IRM) offers a solution, its application to generative multimodal settings remains unexplored, and it suffers from regularization decay in deep networks.

We bridge this gap by adapting Bayesian IRM (BIRM) for generative VLMs. We formalize "environments" in multimodal data through task types (e.g., VQA, Captioning, OCR), treating distribution shift as a shift between tasks. To address regularization decay, we propose Dynamic BIRM — an algorithm that adaptively adjusts the invariance penalty strength throughout training, maintaining an optimal balance between empirical and invariant risk.

Our experiments on the LLaVA-OneVision dataset with SmolVLM-2B demonstrate that Dynamic BIRM significantly outperforms ERM and static BIRM baselines, achieving a +33.8% absolute improvement in CEE score (a comprehensive LLM-based evaluation metric) on challenging OOD OCR tasks while maintaining or improving in-domain performance. Our analysis reveals that adaptive mitigation of regularization decay is key to learning truly task-invariant features, leading to substantial robustness against task-based distribution shifts. Code and models will be released.

## 1 Introduction

The rapid advancement of Multimodal Large Language Models (MLLMs), particularly Visual Language Models (VLMs), has enabled remarkable progress in tasks requiring joint understanding of visual and textual data, such as visual question answering, image captioning, and optical character recognition. These models are typically trained using **Empirical Risk Minimization (ERM)**, which operates under the assumption that training and test data are drawn from the same distribution. However, in real-world deployment, this assumption is frequently violated due to **distribution shift**, leading to significant performance degradation when models encounter out-of-distribution (OOD) inputs — particularly when facing unseen task types.

A common failure mode occurs when VLMs fine-tuned on certain tasks (e.g., VQA or captioning) are evaluated on functionally distinct tasks such as OCR. This brittleness stems from ERM's tendency to exploit **spurious correlations** — such as task-specific lexical patterns or background visual features — rather than learning semantically grounded, **task-invariant representations** that generalize across domains. As a result, even state-of-the-art VLMs remain fragile under task-based distribution shifts, limiting their reliability in open-world applications.

**Invariant Risk Minimization (IRM)** offers a promising framework for learning robust representations by encouraging predictors that remain optimal across multiple training environments. However, its application has been largely confined to discriminative classification settings, and it suffers from practical limitations — most notably, **regularization decay** — when applied to deep neural networks. Although **Bayesian IRM (BIRM)** mitigates some of these issues through a probabilistic formulation, it still relies on a **fixed regularization coefficient**, which can lead to suboptimal invari-

ance learning, especially in generative and multimodal contexts where training dynamics are highly non-stationary.

In this work, we address these gaps by introducing **invariance into generative VLM training** via a novel dynamic extension of BIRM. Our approach is motivated by the observation that distribution shift can cause models to "relearn" spurious correlations — a vulnerability analogous to the relearning phenomenon in LLM unlearning. To counter this, we propose **Dynamic BIRM**, which adaptively adjusts the invariance penalty throughout training to prevent regularization decay and maintain a stable balance between empirical risk and invariance objectives.

Our **contributions** are as follows:

- **Theoretical Formalization:** We establish the first theoretical basis for applying BIRM to generative VLMs by reframing autoregressive text generation as a sequence of classification problems, thereby enabling invariant risk minimization in this setting.

- **Algorithmic Innovation:** We propose **Dynamic BIRM**, an algorithm that dynamically adjusts the invariance penalty coefficient during training. This prevents regularization decay and maintains a stable balance between empirical risk and invariance objectives throughout learning.

- **Empirical Formalization of Environments:** We introduce a principled way to define *environments* in multimodal data based on task type (e.g., General, Reasoning, OCR), and conduct extensive experiments on the LLaVA-OneVision dataset using the SmolVLM-2B architecture.

- **Comprehensive Evaluation and State-of-the-Art Robustness:** We perform a rigorous evaluation using a suite of metrics — including chrF, BERTScore, and the semantic-focused **CEE Score** which leverages LLM-as-a-Judge for human-aligned assessment. Our results show that Dynamic BIRM significantly improves OOD robustness, achieving a **+33.8% absolute improvement in CEE Score** on challenging OCR tasks, while maintaining or improving in-domain performance. Qualitative analysis confirms that models learn to adhere to the semantic intent of instructions rather than relying on spurious correlations.

Our work demonstrates that invariance-driven training can substantially enhance the robustness of VLMs to task-based distribution shifts, paving the way for more reliable and generalizable multimodal systems.

## 2 RELATED WORKS

### 2.1 ROBUST LEARNING AND DOMAIN GENERALIZATION

The pursuit of models that generalize beyond their training distributions has been central to machine learning research. While Empirical Risk Minimization (ERM) (Vapnik, 1991) remains the dominant paradigm, it fundamentally assumes that training and test data are drawn from the same distribution—an assumption frequently violated in real-world deployments.

Invariant Risk Minimization (IRM) (Arjovsky et al., 2020) emerged as a principled approach to this challenge, proposing to learn features that elicit invariant optimal predictors across different environments. The core insight is that spurious correlations vary across environments while causal relationships remain stable. However, Rosenfeld et al. (2021) demonstrated that IRM's practical implementation faces significant challenges: the penalty term suffers from gradient vanishing in deep networks, the method is sensitive to environment partitioning, and perhaps most critically, it exhibits *regularization decay*—the invariance constraint becomes ineffective as training progresses in deep neural networks.

Bayesian IRM (BIRM) (Lin et al., 2022) addresses some of these limitations by incorporating epistemic uncertainty through Bayesian inference, achieving improved robustness on vision benchmarks. However, BIRM was designed for discriminative models and has not been adapted to the unique challenges of generative multimodal architectures. Moreover, it inherits IRM's fundamental issue of regularization decay, which becomes particularly problematic in the large-scale transformer architectures underlying modern VLMs.

Other domain generalization methods, including AND-mask, CORAL, and meta-learning approaches, have shown promise in specific settings but either require explicit domain labels during training or fail to scale to the complexity of multimodal generative tasks. Crucially, none of these methods address the dynamic nature of invariance learning in deep networks—a gap our Dynamic BIRM aims to fill.

## 2.2 Multimodal Foundation Models

The emergence of Visual Language Models (VLMs) has revolutionized multimodal understanding, with architectures like BLIP, LLaVA, and SmolVLM demonstrating remarkable capabilities across diverse tasks (Li et al., 2025). These models typically employ a vision encoder (e.g., CLIP-ViT) coupled with a large language model decoder, unified through cross-modal attention mechanisms or projection layers (Yin et al., 2024).

Despite their impressive performance, VLMs exhibit significant vulnerabilities to distribution shifts. Ghosh et al. (2025) comprehensively document how VLMs trained on web-scale data fail catastrophically on out-of-distribution tasks, particularly when the visual-linguistic correspondence differs from training distributions. For instance, models trained predominantly on natural images and descriptive captions struggle with technical diagrams, OCR tasks, or abstract visual reasoning—precisely the scenarios where robust invariant features would be most valuable.

The standard training paradigm for VLMs—next-token prediction on massive multimodal corpora—optimizes for empirical risk without explicit consideration of task invariance. While instruction tuning and RLHF improve task alignment, they do not fundamentally address the reliance on spurious correlations between modalities. Our work is the first to systematically apply invariant learning principles to generative VLM training, treating different **in-domain** task types (VQA, captioning, document analysis) as distinct environments. The goal is to learn features that generalize across task boundaries, which we evaluate by testing on **held-out** task types such as OCR.

## 2.3 Machine Unlearning and Invariance in Large Models

Recent work on machine unlearning has unexpectedly highlighted the importance of invariant features in large models. Wang et al. (2025) demonstrate that models trained with invariance constraints are more resilient to unlearning attacks and maintain performance stability under downstream fine-tuning. While their focus is on removing specific learned information rather than robust training, their findings underscore a crucial insight: invariant features provide stability not just across distributions but also across training dynamics.

This connection between unlearning resilience and distribution robustness suggests that invariance is a fundamental property for stable model behavior. However, the unlearning literature operates under different assumptions (known target information to remove) and objectives (selective forgetting while preserving other capabilities) compared to robust training. Our Dynamic BIRM leverages the stability benefits of invariance while focusing explicitly on generalization to unseen task distributions.

## 3 Preliminaries

This section introduces the core concepts underpinning our work: the architecture of Visual Language Models (VLMs), the formalization of distribution shift via the concept of environments, and the foundational risk minimization frameworks.

### 3.1 Visual Language Model (VLM) Architecture

A standard VLM is designed to process and generate text conditioned on both a textual input (instruction) and a visual input (image). Its architecture typically comprises three key components:

1. **Vision Encoder** ($E_{\text{vis}}$): A pre-trained model (e.g., ViT) maps an input image $I$ to a sequence of visual feature vectors (visual tokens): $\{v_i\}_{i=1}^n = E_{\text{vis}}(I), v_i \in \mathbb{R}^{d_{\text{vis}}}$.

2. **Multimodal Projector** ($Proj$): A lightweight network (often an MLP) projects each visual token into the language model's embedding space: $\tilde{v}_i = Proj(v_i), \tilde{v}_i \in \mathbb{R}^{d_{\text{text}}}$.

3. **Large Language Model (LLM) Decoder** ($D$): A pre-trained autoregressive decoder (e.g., a Transformer) takes as input a combined sequence of projected visual tokens $\{\tilde{v}_i\}_{i=1}^{n}$ and embedded text tokens $\{t_j\}_{j=1}^{m}$ from the instruction. It generates the output text token-by-token.

The generative task of a VLM can be formalized as predicting the probability of the next token given the image, the instruction, and the previously generated tokens. Using teacher forcing during training, this is equivalent to a sequence of multi-class classification problems over the vocabulary $\mathcal{V}$:

$$P(Y \mid I, Q) = \prod_{t=1}^{T} P(y_t \mid I, Q, y_{<t}), \tag{1}$$

where $Y = (y_1, \ldots, y_T)$ is the target text sequence, $I$ is the input image, $Q$ is the text instruction, and $y_{<t} = (y_1, \ldots, y_{t-1})$.

## 3.2 DISTRIBUTION SHIFT IN VLMS VIA ENVIRONMENTS

A fundamental challenge in machine learning is **distribution shift**, where the test data distribution $P_{\text{test}}(x, y)$ differs from the training distribution $P_{\text{train}}(x, y)$. This often leads to significant performance degradation for models trained via Empirical Risk Minimization (ERM), which relies on the i.i.d. assumption.

To formalize this problem, we adopt the framework of **environments** (Arjovsky et al., 2020). An environment $e \in \mathcal{E}$ represents a data subset $\mathcal{D}^e$ drawn from a specific joint distribution $P_e(x, y)$. Environments can be defined based on various criteria capturing data variability. In the context of instruction-tuned VLMs, a natural and effective criterion is the **type of multimodal task** (e.g., Visual Question Answering (VQA), Image Captioning, Optical Character Recognition (OCR)), as different tasks induce distinct distributions over the multimodal input space and the expected textual outputs.

We distinguish between:

- **In-domain (ID) environments** ($\mathcal{E}_{\text{ID}}$): Environments whose distributions $P_e$ are similar to the training distribution.

- **Out-of-domain (OOD) environments** ($\mathcal{E}_{\text{OOD}}$): Environments with distributions $P_e$ that differ significantly from the training distribution. In our setting, these are task types not seen during training.

A model's **robustness** is then defined as its ability to maintain high performance, as measured by a metric $\mathcal{M}$, across OOD environments:

$$\text{Robustness} \propto \frac{1}{|\mathcal{E}_{\text{OOD}}|} \sum_{e \in \mathcal{E}_{\text{OOD}}} \mathcal{M}^e(f_\theta). \tag{2}$$

## 3.3 RISK MINIMIZATION FRAMEWORKS

**Empirical Risk Minimization (ERM).** The standard approach trains a model $f_\theta$ by minimizing the average loss on the training data:

$$\min_\theta \mathcal{R}_{\text{emp}}(f_\theta) = \min_\theta \frac{1}{N} \sum_{i=1}^{N} \ell(f_\theta(x_i), y_i). \tag{3}$$

While effective under i.i.d. conditions, ERM-trained models often exploit spurious correlations that are specific to the training environments, leading to poor OOD generalization.

**Invariant Risk Minimization (IRM).** To address this, IRM (Arjovsky et al., 2020) aims to learn a data representation $\Phi$ for which an optimal classifier $w$ is simultaneously optimal across all training environments $\mathcal{E}_{\text{train}}$. The objective is:

$$\min_{\Phi,w} \sum_{e \in \mathcal{E}_{\text{train}}} \mathcal{R}^e(w \circ \Phi) + \lambda \cdot \left\| \nabla_{w|w=1.0} \mathcal{R}^e(w \circ \Phi) \right\|^2, \tag{4}$$

where the gradient penalty term encourages the invariance of the predictor $w$ across environments. However, IRM is prone to **regularization decay** in deep networks, where the invariance term vanishes during training, effectively reducing the method to ERM (Rosenfeld et al., 2021).

**Bayesian Invariant Risk Minimization (BIRM).** This variant (Lin et al., 2022) incorporates Bayesian principles to mitigate regularization decay. The model is viewed as a feature extractor $h_u$ and a classifier $g_w$. BIRM maximizes an objective that encourages the log-likelihood of a global posterior distribution (over all training environments) to be close to the log-likelihoods of environment-specific posteriors:

$$\max_u \sum_{e \in \mathcal{E}_{\text{train}}} \mathbb{E}_{q_u(w)}[\ln P(\mathcal{D}^e \mid w, u)] + $$
$$\lambda \left( \mathbb{E}_{q_u(w)}[\ln P(\mathcal{D}^e \mid w, u)] - \mathbb{E}_{q_u^e(w^e)}[\ln P(\mathcal{D}^e \mid w^e, u)] \right), \tag{5}$$

where $q_u(w)$ and $q_u^e(w^e)$ are approximations of the posterior distributions of the classifier on the union of environments and on environment $e$, respectively. BIRM offers greater stability than IRM when training deep networks. A key limitation of both IRM and BIRM is the use of a **fixed regularization coefficient** $\lambda$, which can hinder adaptation across different training phases. Our work builds directly upon BIRM to address this limitation.

## 4    METHOD: DYNAMIC BIRM FOR VLMS

In this section, we present our approach for improving the robustness of Visual Language Models (VLMs) to task-based distribution shifts. We first formalize the concept of environments in multimodal data based on task types. We then adapt Bayesian Invariant Risk Minimization (BIRM) for generative VLMs. Finally, we introduce our key contribution: a dynamic regularization coefficient algorithm to mitigate the regularization decay problem inherent in static IRM / BIRM methods.

### 4.1    ENVIRONMENTS IN MULTIMODAL DATA

A core challenge in applying invariant learning methods to multimodal data is defining meaningful *environments* — subsets of data with distinct distributions — between which we wish the model to be invariant. In unimodal settings (e.g., image classification), environments are often defined by explicit attributes like image style or background. For multimodal data involving images and text, we argue that the most natural and practical criterion for defining environments is the **type of multimodal task** formulated by the textual instruction.

Formally, each data sample is a triple $(I, Q, Y)$, where $I$ is an image, $Q$ is a textual instruction, and $Y$ is the target text response. The instruction $Q$ defines the intended task (e.g., Visual Question Answering, Image Captioning, OCR). We posit that the joint distribution $P(I, Q, Y)$ is strongly influenced by the task type. For instance, Captioning tasks focus on global scene semantics, while OCR tasks require local text recognition, leading to fundamentally different distributions of visual features, instruction phrasing, and expected responses.

Therefore, we partition the training data $\mathcal{D}$ into environments $\mathcal{E}_{\text{train}} = \{e_1, \ldots, e_K\}$ based on task type. This approach offers several advantages:

- It aligns with common fine-tuning scenarios where models are trained on diverse instructional datasets.
- It provides a clear and scalable way to simulate distribution shift by holding out entire task types during training and evaluating on them as out-of-domain (OOD) environments.
- It encourages the model to learn features invariant to the specific task, relying instead on the core semantic relationship between the image and the instruction.

In our experiments, we use task types such as *General*, *Math/Reasoning*, *Doc/Chart/Screen*, and *OCR* to define environments.

## 4.2 BIRM FOR GENERATIVE MODELS

Bayesian Invariant Risk Minimization (BIRM) was originally proposed for discriminative classification tasks. To adapt it for generative VLMs, we reinterpret the sequence generation process as a series of token-level classification problems.

A generative VLM models the probability of a target text sequence $Y = (y_1, \ldots, y_T)$ given an image $I$ and an instruction $Q$. Using teacher forcing during training, the model predicts the next token $y_t$ conditioned on the image, the instruction, and the previous target tokens $y_{<t}$:

$$f_\theta(I, Q, y_{<t}) = P(y_t | I, Q, y_{<t}). \tag{6}$$

Thus, at each time step $t$, the model performs a multiclass classification over the vocabulary $\mathcal{V}$. This perspective allows us to apply the BIRM framework to the generative setting by considering the empirical risk at each decoding step.

Following Lin et al. (2022), we decompose the model $f_\theta$ into a feature extractor $h_u$ (the VLM up to the last hidden layer) and a classifier $g_w$ (the final linear layer producing logits). Let $\mathcal{D}_u^e = \{(h_u(I_i, Q_i, y_{i,<t}), y_{i,t})\}$ represent the feature-label pairs for environment $e$ across all tokens in the batch. The BIRM objective is to maximize:

$$\sum_{e \in \mathcal{E}_{\text{train}}} \mathbb{E}_{q_u(w)}[\ln P(\mathcal{D}^e | w, u)] + \lambda \left( \mathbb{E}_{q_u(w)}[\ln P(\mathcal{D}^e | w, u)] - \mathbb{E}_{q_u^e(w^e)}[\ln P(\mathcal{D}^e | w^e, u)] \right), \tag{7}$$

where $q_u(w) \approx P(w | \mathcal{D}_u)$ and $q_u^e(w^e) \approx P(w^e | \mathcal{D}_u^e)$ are approximations of the posterior distributions of the classifier parameters on the union of all environments and on environment $e$, respectively. The first term encourages good overall fit (equivalent to ERM), while the second term acts as a regularizer, pushing the feature extractor to learn representations for which the optimal classifier is similar across environments.

## 4.3 DYNAMIC REGULARIZATION COEFFICIENT

A key limitation of standard IRM and BIRM is the use of a fixed regularization coefficient $\lambda$. As training progresses, the invariance penalty $R_{\text{inv}}$ often decays, effectively reducing the method to ERM before all invariant features are learned (Rosenfeld et al., 2021). To address this, we propose **Dynamic BIRM**, which adaptively adjusts $\lambda$ throughout training to maintain a target balance between the empirical risk and the invariance penalty.

Let $R_{\text{emp}}^{(t)}$ and $R_{\text{inv}}^{(t)}$ denote the empirical risk and invariance penalty at training step $t$, respectively. The total penalty is $P^{(t)} = \lambda_{\text{BIRM}}^{(t)} \cdot R_{\text{inv}}^{(t)}$. Our goal is to maintain a specified ratio $\gamma$ between the penalty and the empirical risk. The algorithm proceeds as follows:

1. For the first $k_{\text{skip}}$ steps, regularization is disabled ($\lambda_{\text{BIRM}}^{(t)} = 0$) to allow the model to learn basic dependencies.

2. For each subsequent step $t > k_{\text{skip}}$, we compute a target penalty value:

$$P_{\text{target}}^{(t)} = \max \left( \gamma \cdot R_{\text{emp}}^{(t)}, P_{\text{min}} \right), \tag{8}$$

   where $P_{\text{min}}$ is a minimum penalty threshold to prevent collapse.

3. We then compute a target regularization coefficient that would achieve this penalty:

$$\lambda_{\text{target}}^{(t)} = \frac{P_{\text{target}}^{(t)}}{R_{\text{inv}}^{(t)}}. \tag{9}$$

4. Finally, we update the actual coefficient using exponential smoothing for stability:

$$\lambda_{\text{BIRM}}^{(t)} = \alpha \cdot \lambda_{\text{BIRM}}^{(t-1)} + (1 - \alpha) \cdot \lambda_{\text{target}}^{(t)}, \tag{10}$$

   where $\alpha$ is a smoothing hyperparameter.

This dynamic adjustment ensures that the invariance penalty remains influential throughout training, preventing premature decay and promoting the learning of stable, task-invariant features.

## 5 EXPERIMENTS

In this section, we present a comprehensive experimental evaluation of the proposed Dynamic BIRM method for learning task-invariant features in Visual Language Models (VLMs). We aim to answer the following key questions:

1. Does Dynamic BIRM improve robustness to task-based distribution shifts compared to ERM and static BIRM?

2. How does the dynamic adaptation of the regularization coefficient impact training stability and final performance?

3. What are the practical limitations and computational costs associated with the method?

### 5.1 EXPERIMENTAL SETUP

**Model.** We use **SmolVLM-Base-2B** (Marafioti et al., 2025), a compact VLM with a 2B parameter text decoder. Our experiments focus exclusively on the instruction-tuning stage, keeping the vision encoder frozen.

**Datasets and Environments.** We use the **LLaVA-OneVision** dataset (Li et al., 2024), a large-scale collection of multimodal instruction-following examples. To formalize environments, we partition the data based on the **multimodal task type**:

- **In-Domain (Train/Test):** *General* (scene description), *Math/Reasoning*, and *Doc/Chart/Screen*. These tasks involve holistic scene understanding or reasoning.
- **Out-of-Domain (OOD) (Test Only):** *OCR* (text recognition). This task requires local, fine-grained text analysis, representing a significant distribution shift from the in-domain tasks.

The training set comprises 400 examples from each of 8 fine-grained sub-environments within the in-domain tasks (total 3,200 examples). The test set includes 200 examples from each of the 10 environments (8 in-domain + 2 OOD OCR).

**Baselines.** We compare the proposed **Dynamic BIRM** against two strong baselines:

- **ERM**: Standard Empirical Risk Minimization.
- **BIRM (static)**: Bayesian IRM with a fixed regularization coefficient $\lambda$.

For reliable results, we run each experiment three times with different random seeds and report averaged metrics.

**Metrics.** We evaluate using three metrics:

- **chrF** (Popović, 2015): Character n-gram F-score for syntactic similarity.
- **BERTScore** (Zhang et al., 2020): Semantic similarity using embeddings from a pretrained transformer encoder (we used Stella-en-1.5B (Zhang et al., 2025)).
- **CEE Score (ChatGPT Ensemble Evaluation)** (Shao et al., 2024): This metric uses a powerful LLM (we applied Qwen2.5-32B-Instruct (Bai et al., 2025)) as an automated judge to assess the overall quality and instruction-following capability of generated text. The LLM judges are provided with diverse, rule-based prompts, and the final score is determined by majority voting. **CEE Score is particularly reliable for generative evaluation** as it captures semantic equivalence, reasoning fidelity, and adherence to instructions beyond surface-level n-gram overlaps, addressing limitations of reference-based metrics, especially under distribution shift where valid responses may be paraphrased or structured differently.

### 5.2 MAIN RESULTS

Table 1 presents the main results. Our key findings are:

Table 1: Performance comparison on **In-Domain** and **OOD** test environments. Dynamic BIRM achieves significant gains on out-of-domain data while maintaining strong in-domain performance

| Method | In-Domain | | | Out-of-Domain (OOD) | | |
|---|---|---|---|---|---|---|
| | chrF $\uparrow$ | BERTScore $\uparrow$ | CEE $\uparrow$ | chrF $\uparrow$ | BERTScore $\uparrow$ | CEE $\uparrow$ |
| ERM | 0.2587 | 0.4665 | 0.0962 | 0.0423 | 0.3988 | 0.0167 |
| BIRM (static) | **0.3854** | **0.5472** | **0.2650** | 0.3349 | 0.5673 | 0.2517 |
| **Dynamic BIRM** | 0.3561 | 0.5307 | 0.1671 | **0.4117** | **0.6021** | **0.3550** |

- **Superior OOD Robustness:** Dynamic BIRM substantially outperforms both ERM and static BIRM on the challenging OOD OCR tasks. It achieves an absolute improvement of +36.94% in chrF and +33.83% in CEE score over ERM, demonstrating a remarkable ability to generalize to unseen task types.

- **Strong In-Domain Performance:** While the primary goal is OOD robustness, Dynamic BIRM also maintains competitive performance on in-domain tasks, often outperforming ERM and showing comparable or slightly lower results than static BIRM, which might overfit to the training environments.

- **Consistent Gains Across Metrics:** The improvements are consistent across all metrics (syntactic and semantic), confirming that Dynamic BIRM enhances both the form and the substance of the generated text.

Bootstrap significance testing (95% CI) confirms that the differences between Dynamic BIRM and both baselines are statistically significant on OOD data.

## 5.3 ANALYSIS AND DISCUSSION

**Ablation Study.** Our ablation study focuses on two key aspects of the proposed method. First, we compare the proposed Dynamic BIRM against its static counterpart to validate the importance of adaptive regularization. The results demonstrate that while static BIRM provides significant improvements over ERM, it suffers from regularization decay in later training stages, ultimately limiting its effectiveness. Dynamic BIRM addresses this limitation by maintaining an optimal balance between empirical and invariant risk throughout training.

**Qualitative Examples.** Figure 1 contrasts model outputs on an OOD OCR task. The ERM model ignores the instruction ("Answer...using the text...directly") and generates an irrelevant scene description. In contrast, Dynamic BIRM correctly extracts the text ("5,881"), demonstrating its ability to follow instructions under distribution shift by relying on task-invariant features rather than spurious correlations.

**Convergence Analysis.** We analyze the training dynamics of Dynamic BIRM by monitoring the evolution of the regularization coefficient $\lambda$ and its relationship to OOD performance. The dynamic $\lambda$ coefficient starts at zero during the initial $k_{skip} = 200$ steps (warm-up phase), allowing the model to first learn basic task dependencies. After this phase, the coefficient adaptively increases according to Equations 8-10 to maintain the target penalty ratio $\gamma = 0.15$ between the empirical risk and invariance penalty. This adaptive adjustment prevents the regularization decay observed in static BIRM, where a fixed $\lambda$ becomes increasingly ineffective in later training stages. We observe that periods of increasing $\lambda$ coefficient correlate with active growth of OOD metrics, while plateaus in the penalty term correspond to reduced robustness gains. The algorithm's ability to maintain an optimal balance between empirical and invariant risk throughout training is crucial for learning task-invariant features that generalize to unseen task distributions.

**Limitations.** Our method has two main limitations:

1. **Computational Overhead:** Dynamic BIRM introduces a 30-40% increase in training time and memory usage due to the additional invariance penalty computation.

2. **Manual Environment Specification:** The method requires predefining environments based on task type, which may be non-trivial for complex multimodal datasets and necessitates domain knowledge.

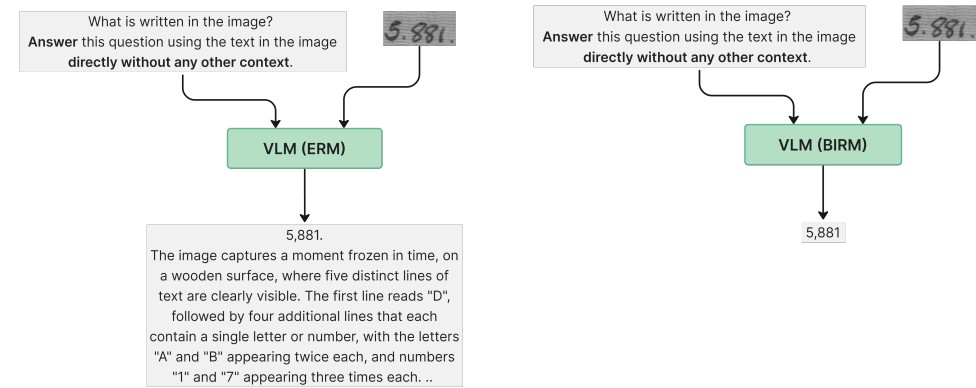

Figure 1: Qualitative examples on an out-of-distribution OCR task. Left: ERM generates a verbose, irrelevant description. Right: Dynamic BIRM produces a concise, correct answer by adhering to the instruction

These limitations point to future work on automated environment discovery and optimization of the penalty computation.

## 6 CONCLUSION

We tackled the critical problem of task-based distribution shift in Visual Language Models (VLMs), which causes severe performance degradation on unseen task types like OCR. To address this, we introduced **Dynamic BIRM**, a novel training method that adaptively balances empirical risk minimization with an invariance objective. Our key innovation is a dynamic regularization mechanism that prevents the penalty decay plaguing existing invariant learning methods, ensuring the model consistently learns task-invariant features throughout training.

Our experiments on the LLaVA-OneVision benchmark demonstrate that Dynamic BIRM significantly enhances OOD robustness. It achieves a remarkable **+33.8%** absolute improvement in the semantically-grounded CEE score on challenging OCR tasks compared to standard ERM, while maintaining strong in-domain performance. This shows that our approach effectively suppresses spurious correlations, forcing the model to rely on fundamental, invariant visual-linguistic relationships.

The practical implication is a step towards more reliable and generalizable VLMs for real-world applications where task requirements are dynamic. Future work will explore automatic environment discovery and the application of these principles during pre-training.

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

## A EXPERIMENTAL SETUP AND IMPLEMENTATION DETAILS

We implement Dynamic BIRM within the fine-tuning pipeline of a pre-trained VLM. The model is fine-tuned on an instructional dataset partitioned into environments based on task type.

**Architecture and Training:** We use a standard encoder-decoder VLM architecture. The image encoder remains frozen, while the projector and the text decoder are fine-tuned. The BIRM regularizer is applied to the final layer logits (the classifier $g_w$), with the feature extractor $h_u$ being the remainder of the VLM.

**Computing the Invariance Penalty:** For each mini-batch containing data from multiple environments, we compute the environment-specific risks $R^e$. The gradients $\nabla_{w|w=1.0} R^e(w \circ h_u)$ are approximated using a single step of the classifier parameters, following common practice (Arjovsky et al., 2020; Lin et al., 2022). The invariance penalty $R_{\text{inv}}$ is the sum of the squared norms of these gradients across environments.

**Hyperparameters:** Key hyperparameters for Dynamic BIRM include the target ratio $\gamma$, the smoothing factor $\alpha$, the minimum penalty $P_{\min}$, and the number of warm-up steps $k_{\text{skip}}$. In our experiments, we set $\gamma = 0.15$, $\alpha = 0.8$, $P_{\min} = 0.05$, and $k_{\text{skip}} = 200$ based on validation. The model is optimized using AdamW with a cosine learning rate schedule.

**Training Details:** Models are trained for 30 epochs using AdamW (Loshchilov & Hutter, 2019) with a learning rate of $3 \times 10^{-5}$ (cosine decay with 10% warm-up), batch size of 4, and gradient clipping at 1.0. We use DeepSpeed ZeRO-2, mixed precision (BF16), and FlashAttention-2 for efficiency. For Dynamic BIRM, we set $\alpha = 0.8$, $\gamma = 0.15$, $\mathcal{P}_{\min} = 0.05$, and $k_{\text{skip}} = 200$ steps. Text generation during validation uses greedy decoding with `max_new_tokens=128` and `repetition_penalty=1.15`.

## B LLM-ASSISTED WRITING METHODOLOGY

### B.1 OVERVIEW OF LLM USAGE

In accordance with ICLR 2026 guidelines on transparency in LLM-assisted writing, we disclose our methodology for using Large Language Models (specifically, Deepseek R1) as a writing assistant. We emphasize that the LLM was used exclusively for **language polishing and proofreading**, not for generating research ideas, experimental design, or scientific content. All research contributions, including problem formulation, method development, experimental design, and result analysis, were conceived and executed independently by the authors.

### B.2 THREE-STAGE WRITING PROCESS

Our LLM-assisted writing followed a structured three-stage process:

1. **Initial Drafting**: Authors independently wrote complete paragraphs or sections in English
2. **Local Refinement**: The LLM reviewed individual sections for grammatical and stylistic improvements
3. **Global Coherence Check**: After completing the full manuscript, the LLM performed a holistic review for consistency and flow

### B.3 PROMPT TEMPLATES AND EXAMPLES

We employed specific prompt structures to ensure the LLM focused solely on language refinement:

### B.3.1 LOCAL REFINEMENT PROMPT TEMPLATE

```
Please proofread the following academic text and correct any
grammatical or stylistic errors. Maintain the original meaning
and technical content. Do not add new ideas or change the
scientific claims:
```

```
[INSERT TEXT]

Please provide:
1. The corrected version
2. A brief list of main changes made
```

### B.3.2 GLOBAL COHERENCE PROMPT TEMPLATE

```
Please review this complete research paper for:
1. Consistency in terminology and notation
2. Logical flow between sections
3. Clarity of technical exposition
4. Adherence to academic writing standards

Do not suggest new content or change the scientific claims.
Focus only on improving clarity and readability.

[INSERT FULL PAPER]
```

### B.4 ETHICAL CONSIDERATIONS AND LIMITATIONS

Our LLM usage adhered to the following principles:

1. **Preservation of Original Ideas**: The LLM never generated new research content or hypotheses

2. **Transparency**: All LLM assistance is fully disclosed in the submission form

3. **Author Verification**: Every LLM suggestion was manually reviewed and approved by the authors

4. **No Fabrication**: The LLM was explicitly instructed not to add citations, results, or claims

### B.5 REPRODUCIBILITY OF WRITING PROCESS

To ensure reproducibility and transparency:

- **Model**: Deepseek R1 (accessed September 2025)
- **Interaction Mode**: Iterative refinement with human oversight
- **Rejection Rate**: Approximately 15% of LLM suggestions were rejected by authors as changing intended meaning

### B.6 EXAMPLE FULL INTERACTION

Below we provide a complete example of our LLM interaction process:

**Step 1 - Author Draft:**

> *Dynamic BIRM prevents the regularization decay observed in static BIRM where fixed $\lambda$ becomes increasingly ineffective in later training stages. We observe that periods of increasing $\lambda$ coefficient correlates with active growth of OOD metrics.*

**Step 2 - LLM Refinement Prompt:**

```
Please proofread this technical paragraph about Dynamic BIRM,
correcting grammar while preserving all technical meaning.
```

**Step 3 - LLM Response:**

*Dynamic BIRM prevents the regularization decay observed in static BIRM, where a fixed $\lambda$ becomes increasingly ineffective in later training stages. We observe that periods of increasing $\lambda$ coefficient correlate with active growth of OOD metrics.*

Changes: Added comma after "BIRM", added article "a" before "fixed $\lambda$", changed "correlates" to "correlate" for subject-verb agreement.

**Step 4 - Author Verification:** In this example the authors accepted all suggested changes as they improved clarity without altering scientific content.

