# OpenReview forum: "Learning Task-Invariant Features in VLMs via Dynamic Bayesian IRM"
_ICLR.cc/2026/Conference — Submitted to ICLR 2026_

### Official Review · Reviewer_UR8U · 2025-10-27

**Soundness:** 2
**Presentation:** 2
**Contribution:** 1
**Rating:** 2
**Confidence:** 3

**Summary:**

This paper applies Bayesian Invariant Risk Minimization (BIRM) to address the performance degradation of vision-language models (VLMs) in out-of-distribution (OOD) tasks. By introducing a dynamic regularization weight into the original BIRM loss, the proposed method—Dynamic BIRM—enhances model robustness under OOD conditions compared to baseline BIRM approaches.

**Strengths:**

1. Improving the robustness of VLMs is an important and timely research direction. This paper presents a solid case study demonstrating the application of BIRM within the VLM context.

**Weaknesses:**

1. Since the motivation of this work lies in dynamically adjusting the regularization term, it would be beneficial to include empirical analyses showing how the decay of the regularization weight affects model performance.

2. Sections 4.1 and 4.2 both serve as preliminaries; it would improve readability and narrative flow to move these into Section 3.

3. The experimental scope appears limited. Given that the paper’s primary contribution is to improve VLM performance on OOD tasks, evaluating only an OCR dataset is insufficient. It would strengthen the work to include additional OOD benchmarks—such as medical QA, chemical reasoning, or other domain-specific tasks. Furthermore, the authors could consider generating synthetic OOD tasks by applying perturbations to visual or textual inputs to better validate the robustness claims.

**Questions:**

1. In Section 3.3 (Preliminary on BIRM), the paper states that the model can be represented by $h_u$ and $g_w$. However, Equation (5) does not clearly incorporate these terms. Please clarify how $h_u$ and $g_w$ are used within the objective function and their connection to Equation (5).
2. The experimental setup is unclear. Please explicitly specify which datasets or benchmarks were used for evaluation, and clearly distinguish between in-distribution (ID) and out-of-distribution (OOD) tasks.

3. Dynamic BIRM consistently underperforms on ID tasks. Could the authors provide an explanation or discussion of this trade-off between ID and OOD performance?

---

### Official Review · Reviewer_Lt77 · 2025-10-31

**Soundness:** 1
**Presentation:** 2
**Contribution:** 1
**Rating:** 2
**Confidence:** 4

**Summary:**

The paper proposes Dynamic BIRM, an adaptive variant of Bayesian Invariant Risk Minimization for generative VLMs, where “environments” are defined by task types (e.g., VQA, captioning, OCR). It claims improved OOD robustness on a small LLaVA-OneVision split using SmolVLM-2B, chiefly on OCR, by dynamically tuning the invariance penalty during training.

**Strengths:**

1. The paper targets a timely and meaningful goal where task-invariant representations for VLMs under task-based distribution shift.
2. Framing VLM generation as token-level classification to port BIRM and proposing a dynamic penalty schedule is a reasonable direction with potentially useful intuition.

**Weaknesses:**

1.. Introduction lacks citations, leaving prior work and novelty unclear.
2. Narrative redundancy with insufficient exposition of the proposed method.
3. Citation practices are non-standard and inconsistent.
4. Experiments are insufficient: too few baselines, limited models, and narrow evaluation.

**Questions:**

1. Core strands in DG/IRM and multimodal robustness aren’t cited where claims are made, making it hard to judge incremental contribution or distinguish from standard baselines. There is no citation in the whole Introduction part and the whole paper seems use LLM to polish since the abbreviations are redefined (e.g., VLM, ERM).
2. The method section gives a verbal, four-step schedule for $\lambda$ (Eq. 8–10) but no pseudo-code and algorithm box, complexity, or stability analysis beyond prose; definitions of $q_u$, $q_e$ are stated but their approximations are not concretely specified.
3. Mixed formatting, uneven venue info, and in-text citation gaps reduce credibility; the bibliography should follow a consistent style and anchor each claim to canonical prior work. For example, in section 2.2, “Ghosh et al. (2025)” and in section 2.3, “Wang et al. (2025)”
4. Omits standard DG baselines, evaluates mainly on a single small model and narrow OOD setting, and lacks robust ablations or stronger evaluations. For example, baselines are only ERM and static BIRM, but no standard DG competitors, limiting the strength of the claim. Moreover, evaluation uses a single small model (SmolVLM-Base-2B) and defines OOD solely as OCR; the train/test sizes are very small (3,200 train), restricting generality.

---

### Official Review · Reviewer_yzZ3 · 2025-11-01

**Soundness:** 2
**Presentation:** 3
**Contribution:** 2
**Rating:** 2
**Confidence:** 4

**Summary:**

This paper addresses distribution shift in Visual Language Models (VLMs), where traditional ERM fails to learn task-invariant features. The authors adapt Bayesian IRM for generative VLMs and propose Dynamic BIRM, which dynamically adjusts the invariance penalty during training. Experiments on LLaVA-OneVision show substantial improvements on out-of-distribution tasks, particularly OCR, while maintaining or enhancing in-domain performance.

**Strengths:**

1. Introducing invariant learning methods to address out-of-distribution generalization challenges in large generative VLMs is an interesting and promising direction.

2. The paper is well organized and clearly written, making it easy to follow and understand.

**Weaknesses:**

1. The contribution of this work appears somewhat limited. The core novelty lies in introducing Bayesian Invariant Risk Minimization (BIRM) to generative VLMs and designing a dynamic balancing weight for the BIRM objective.

2. There exist many methods in the invariant learning literature (e.g., IRM-IB [1], MRI [2], [3]). The rationale for specifically adopting Bayesian IRM for generative VLMs requires further clarification. Including comparisons with other invariant learning approaches could strengthen the evaluation and highlight the advantages of the proposed method.

3. The proposed method is evaluated on only one dataset and one VLM model. Evaluating it across additional datasets and models (e.g., Qwen3-VL) would enhance the robustness of the results and better demonstrate the general effectiveness of the approach.

4. Only a single out-of-distribution setup (OCR task as OOD, others as in-distribution) is considered. Evaluating additional OOD scenarios (e.g., VQA task as OOD while others are ID) would provide a more comprehensive assessment of the method’s generalization capabilities.

[1] Ahuja, K., Caballero, E., Zhang, D., Gagnon-Audet, J. C., Bengio, Y., Mitliagkas, I., & Rish, I. (2021). Invariance principle meets information bottleneck for out-of-distribution generalization. Advances in Neural Information Processing Systems, 34, 3438-3450.

[2] Huh, D., & Baidya, A. (2022). The Missing Invariance Principle found--the Reciprocal Twin of Invariant Risk Minimization. Advances in Neural Information Processing Systems, 35, 23023-23035.

[3] Montasser, O., Shao, H., & Abbe, E. (2024). Transformation-invariant learning and theoretical guarantees for OOD generalization. Advances in Neural Information Processing Systems, 37, 108649-108673.

**Questions:**

No.

---

### Official Review · Reviewer_w5F9 · 2025-11-07

**Soundness:** 2
**Presentation:** 3
**Contribution:** 2
**Rating:** 4
**Confidence:** 4

**Summary:**

This paper addresses the important problem of distribution shift in Visual Language Models (VLMs), where models fail when encountering out-of-distribution task types not seen during training. The authors propose Dynamic BIRM, which extends Bayesian Invariant Risk Minimization to generative VLMs by (1) reframing autoregressive generation as sequential classification, (2) defining environments based on task types, and (3) introducing a dynamic regularization coefficient that adaptively adjusts during training to prevent regularization decay. Experiments on LLaVA-OneVision with SmolVLM-2B show +33.8% absolute improvement in CEE Score on OOD OCR tasks while maintaining in-domain performance.

**Strengths:**

- Task-based distribution shift is a real challenge for deployed VLMs
- First to adapt Bayesian IRM to generative multimodal models
- +33.8% improvement in CEE Score is substantial

**Weaknesses:**

- No comparison to CORAL, meta-learning, or other domain generalization methods
- No systematic study of hyperparameter sensitivity or alternative design choices
- Lacks formal justification for task-based environments and when dynamic adjustment helps
- 30-40% overhead is significant with no cost-benefit analysis

**Questions:**

- Why only OCR for OOD evaluation? Have you tested on other held-out task types (e.g., mathematical reasoning, chart understanding)?
- How does performance scale with dataset size? Your training set is quite small - do gains persist with larger datasets?
- What about automatic environment discovery? Can task types be discovered automatically rather than manually specified?

---

### Meta-Review · Area_Chair_QTPN · 2025-12-26

**Summary:**

This paper proposes a method to address the distribution shift problem in VLMs. The proposed method, named Dynamic Bayesian Invariant Risk Minimization (Dynamic BIRM), is an adaptive variant of BIRM for generative VLMs. "Environments" are defined by task types, such as, VQA or captioning. Experiment results are shown in LLaVA-OneVision with SmolVLM-2B.

**Reviewer Concerns:**

- The evaluation result is limited to a single model and a single OCR task (w5F9, yzZ3, Lt77, UR8U)
- Lack of comparisons with state-of-the-art domain generalization (DG), IRM methods, or meta-learning methods (w5F9, yzZ3, Lt77)
- Lack of technical novelty. The core contribution is from BIRM (yzZ3)
- No hyperparameter study or design choice study (this is a critical problem as pointed out by the DomainBed paper) (w5F9)
- Significant computational overhead (30-40%) (w5F9)
- Lack of justification for the design choice, such as task-based environments, or dynamic adjustment (w5F9, UR8U)
- Writing concerns (Lt77, UR8U)

**Reviewer Scores:**

The reviewers reached a negative consensus, while no rebuttal was provided during the discussion period. I agree with the reviewers' opinion.

---

### Decision · Program_Chairs · 2026-01-26

Reject